# Workload-Related Issues among Nurses Caring for Patients with Behavioral and Psychological Symptoms of Dementia: A Scoping Review

**DOI:** 10.3390/healthcare12181893

**Published:** 2024-09-21

**Authors:** Younhee Kang, Chohee Bang

**Affiliations:** 1College of Nursing, Ewha Womans University, Seoul 03760, Republic of Korea; 2Graduate Program in System Health Science and Engineering, Ewha Womans University, Seoul 03760, Republic of Korea; 3Department of Nursing, College of Health Science, Honam University, Gwangju 62399, Republic of Korea

**Keywords:** dementia, nurses, behavioral and psychological symptoms of dementia (BPSD), workload, work-related stress, scoping review

## Abstract

Background/Objectives: As the elderly population grows, the prevalence of dementia is rising, with 70–95% of patients in hospital settings exhibiting problematic behaviors such as aggression. These behaviors significantly contribute to increased nursing workloads, affecting nurses’ well-being and patient care quality. This study aims to review workload-related issues among nurses caring for dementia patients, highlighting the need for targeted interventions to mitigate stress and improve care quality. Methods: A scoping review was conducted using the five-stage framework of Arksey and O’Malley. The literature search covered studies published between 2013 and 2023, focusing on quantitative research about nurses’ workload-related stress when managing patients with dementia and problematic behaviors. Databases such as PubMed and PsycINFO were searched, and 13 studies were selected based on predefined inclusion and exclusion criteria. Results: The review revealed that problematic behaviors, particularly aggression, significantly increase nurses’ stress and workload. This stress has negative consequences on nurses’ physical and mental health, often leading to burnout, decreased job satisfaction, and a decline in care quality. Inadequate staffing and support systems exacerbate these issues. Conclusions: Targeted education, sufficient staffing, and support are essential to reduce the workload and stress experienced by nurses caring for dementia patients. Implementing these strategies can enhance the quality of care provided and improve the well-being of healthcare professionals.

## 1. Introduction

Currently, more than 55 million people globally live with dementia, and this number is expected to exceed 131 million by 2050 [1]. The prevalence of dementia is increasing worldwide as the population ages. The number of people with dementia in South Korea is 840,000 and is projected to reach 2.71 million by 2050 [2]. Dementia is a syndrome characterized by chronic or progressive brain disease, difficulties with short-term memory, and cognitive deficits [1]. It affects cognitive function and leads to problematic behaviors, with half of all patients with dementia experiencing at least one behavioral psychological symptom of dementia (BPSD) [3].

Patients with dementia encountered in hospitals are more likely to exhibit problematic behaviors owing to environmental changes [4]. This factor is especially true for patients with dementia in acute care setting, where the severity of problematic behaviors can be more pronounced [5]. Problematic behaviors of patients with dementia increase the workload and stress experienced by nurses who are already under pressure from insufficient resources and heavy nursing duties [6].

With increased life expectancy and the rising number of patients with dementia [2], the care of these patients, traditionally handled within families, is shifting to social care institutions such as elder care facilities [7]. Nurses in clinical settings consequently have more opportunities to care for patients with dementia and encounter problematic behaviors. Approximately 78.9% of residents in elder care facilities have dementia [8], and the incidence of problematic behaviors in hospitals due to factors such as environmental changes is higher than in homes [9].

Nurses are required to provide a wide range of nursing services, from basic to advanced care, which makes caring for patients with dementia challenging [10]. In particular, the unpredictable and uncontrollable problematic behaviors of dementia patients place an additional burden on nurses, negatively impacting their physical and mental health and increasing the risk of stress-related illnesses [11,12]. As a result, this leads to nurse burnout, job dissatisfaction, retirement, and increased social costs [1,10,13]. In the process of nursing care for dementia patients exhibiting problematic behaviors, nurses experience significant stress and communication difficulties, which leads to emotional labor [14]. High emotional labor and stress levels significantly lower nurses’ quality of life [15] and negatively affect the quality of the nursing care provided [16]. The unpredictable problematic behaviors of dementia patients, such as violence and wandering, cause substantial stress for nurses, leading to burnout [17]. Burnout is conceptualized as a form of workplace stress characterized by exhaustion, mental detachment, and difficulties in both cognitive and emotional functioning [18]. Consequently, nurses struggle to provide appropriate care, which increases the risk of cardiovascular and mental health issues [19], reduces job satisfaction, and lowers the quality of nursing care [20].

Finally, nurses caring for dementia patients experience high levels of subjective stress, which negatively impacts their overall well-being, promoting depression and affecting their mental health and quality of life, thus resulting in health deterioration [21]. Despite the significant burden and challenges faced by nurses, most studies on this topic are limited to qualitative research on caregivers (e.g., family, etc.), with quantitative studies being scarce and lacking integration. Therefore, reviewing relevant research is necessary to understand the trends in work-related stress among nurses caring for dementia patients, suggest future research directions, and propose effective interventions to reduce their work-related stress.

The first aim of this study is to identify research trends in workload-related issues among nurses caring for patients with dementia exhibiting problematic behaviors via a scoping review. By analyzing and organizing the existing research, the second aim of this study is to suggest future research directions and provide guidance for reducing nurses’ workload and delivering effective nursing interventions.

The research question is the following: what is the extent of workload-related stress experienced by nurses caring for behavioral and psychological symptoms of dementia and what measurement tools are used?

## 2. Materials and Methods

This study is a scoping review aimed at exploring workload-related issues faced by nurses caring for dementia patients who exhibit problematic behaviors. The primary objectives of the literature search were to identify research trends, suggest future research directions, and provide evidence-based guidance for reducing nurses’ workload while delivering effective nursing interventions.

The scoping review followed the methodological framework for scoping studies developed by Arksey and O’Malley [22]. This framework is particularly suited for mapping the available evidence, identifying knowledge gaps, clarifying key concepts, and examining how previous studies on related topics were conducted [23]. Scoping reviews allow for the inclusion of a wide variety of research designs, which enables the mapping and summarizing of the existing research on the subject [22]. In this study, the scoping review method was deemed appropriate for examining factors related to workload issues among nurses caring for dementia patients with problematic behaviors.

Scoping reviews, as a rule, do not formally assess the quality or relative strength of the evidence included [22]. Consequently, no critical appraisal of the studies was performed in this review. The five steps of the scoping review framework are outlined below: (1) identifying the research question; (2) identifying relevant studies; (3) selecting studies; (4) charting the data; (5) collating, summarizing, and reporting the results. The Methods section provides detailed descriptions of stages 2 to 4, while the Results section focuses on stage 5. Recent updates to scoping review methodology were incorporated to enhance the review process [22].

Additionally, this study adhered to the PRISMA (Preferred Reporting Items for Systematic Reviews and Meta-Analyses extension for Scoping Reviews; PRISMA-ScR) guidelines (Appendix A). No formal review protocol was written or registered with any scoping review registry.

### 2.1. Identifying the Research Question

The aim of the research question was to analyze the workload-related stress and measurement variables for nurses caring for patients with dementia exhibiting problematic behaviors. The research question was the following: what is the extent of workload-related stress experienced by nurses caring for behavioral and psychological symptoms of dementia and what measurement tools are used?

### 2.2. Identifying Relevant Studies

The literature search and selection processes followed the Preferred Reporting Items for PRISMA (Preferred Reporting Items for Scoping Reviews and Meta-Analyses extension for Scoping Reviews; PRISMA-ScR) guidelines [24]. 

In the past decade, the rise in dementia cases due to aging has prompted countries to enhance public health responses [1]. During this time, the role of nurses has evolved, and technological advances have improved patient care efficiency. A 10-year period was chosen for the literature review to capture recent trends in policies, technology, and demographics, ensuring a comprehensive reflection of the latest developments in dementia care [25]. 

The search targeted studies published from 1 January 2013 to 31 December 2023 that focused on the following topics: (1) nurses caring for patients with dementia and problematic behaviors; (2) stress; (3) quantitative research. Only papers published in English or Korean in domestic and international journals were included. Qualitative studies, dissertations, reports, conference abstracts, and unpublished literature were excluded. The search used domestic and international electronic databases, including PubMed, Cumulative Index to Nursing and Allied Health Literature, Web of Science, and PsycINFO (ProQuest) for international studies and Research Information Sharing Service, Korean Studies Information Service System, and Korean Database of Periodical Information Academic and the National Assembly Library for domestic studies. Keywords used were “Dementia”, “Problematic behaviors”, “Behavioral and psychological symptoms of dementia, BPSD”, “Nurs*”, “stress”, “work*”, and “workload”, combined with the “AND” operator.

### 2.3. Study Selection

Following the recommendations for scoping reviews, two reviewers independently screened and selected the studies, with any discrepancies resolved via discussion or by a third reviewer if a consensus could not be reached. The initial search yielded 3167 articles, with 1080 duplicates removed, leaving 2087 articles. The titles and abstracts of these articles were screened, and 37 articles were selected for full-text review. After reviewing the full texts, 24 articles were excluded because they did not meet the inclusion criteria, such as studies not being relevant to the topic or studies focusing on subjects other than registered nurses (e.g., social workers or certified nursing assistants). Thirteen articles were ultimately included in this review. The selection process is illustrated in Figure 1. The two reviewers achieved a consensus without requiring a third reviewer.

### 2.4. Charting the Data

Data extraction followed the scope review recommendations of Arksey and O’Malley’s [22]. This stage involved extracting summaries from each paper by data extraction. Key information from the selected studies, including author(s), year of publication, research purpose, study population, study location, research methods, measurement variables, study results, outcome variables, and main findings, were extracted and tabulated.

## 3. Results

### 3.1. General Characteristics of the Literature

The selected articles [26,27,28,29,30,31,32,33,34,35,36,37,38] are shown in Table 1, which provides a summary of the studies. From 2013 to 2023, 13 studies focused on workload-related stress in nurses caring for patients with dementia and problematic behaviors. The highest number of studies (four studies) was published in 2018 (Table 1). Regarding the research methods, 12 (84.6%) of the 13 studies were cross-sectional, whereas 2 (15.4%) studies were quasi-experimental (Table 2). Geographically, the studies were conducted in South Korea (three studies, 23.1%), Japan (two studies, 15.4%), and Jordan, Germany, the Netherlands, Canada, New Zealand, Switzerland, Spain, and the United States (one study each). The study revealed that over one-third (38.5%) of the total literature originated from research conducted in Asian countries.

### 3.2. Research Instruments

The workload-related stress of nurses caring for patients with dementia and problematic behaviors was measured using stress measurement tools [27,28,30,31,32,33,35] and burden [26,34,37,38] and job-related quality of life assessment tools [29,37]. More than half of the studies (53.8%), specifically 7 out of 13, measured work-related stress using stress measurement tools. Stress was assessed using seven tools: the Brief Job Stress Questionnaire, Holmes and Rahe Stress Scale, Kessler Psychological Distress Scale, Neuropsychiatric Inventory, Perceived Neighborhood Justice Scale, and Stressful Life Events Screening Questionnaire. The most frequently used tool was the Neuropsychiatric Inventory Questionnaire (NPI-Q), which was used in three studies to measure nurses’ work-related distress [28,31,33]. The NPI was developed to evaluate behavioral disturbances in patients with dementia by assessing the severity of 12 problematic behaviors: delusions, hallucinations, agitation/aggression, depression/dysphoria, anxiety, euphoria/elation, apathy/indifference, disinhibition, irritability/lability, aberrant motor behavior, night-time behavior, and appetite/eating changes. It also measures caregivers’ stress levels and has been widely used for a long time in many studies to assess behavioral disturbances in dementia patients [39].

Five studies measuring workload burden used the Perceived Cognitive Workload Index, Multiple-Method Cognitive Assessment System, Questionnaire for Workplace Capacity, Zarit Burden Interview, and Care Burden of Dementia Disturbing Behavior Scale. Two studies evaluated work-related quality of life by using the Professional Quality of Life Scale and the Nurses’ Working Life Questionnaire to assess burnout and the extent of nurses’ stress. The tools used to measure nurses’ stress in the 13 studies are listed in Table 3.

### 3.3. Study Findings

The analysis of the 13 studies revealed that nurses caring for patients with dementia exhibiting problematic behaviors experienced high levels of workload-related stress. All problematic behaviors of patients with dementia induced stress in nurses, with aggression particularly noted as the primary factor contributing to subjective and objective stress [26,27,28,29,30,31,32,33,34,35,36,37,38]. Additionally, deterioration in activities of daily living objectively increased the nursing burden [34] and led to issues related to insufficient nursing time [37]. The frequency and severity of problematic behaviors also emerged as significant predictors of nurses’ workload-related stress [33,36,37,38]. Such stress negatively affects nursing performance and is associated with increased turnover rates [34]. Furthermore, job-related quality of life, measured in terms of compassion satisfaction, burnout, and compassion fatigue, indicated that compassion satisfaction refers to the pleasure derived from effectively performing one’s job, while burnout denotes feelings of hopelessness and difficulty in coping with work or in performing it effectively, often linked with trauma. Compassion fatigue refers to the secondary exposure to extremely stressful workplaces [38].

Workload-related stress among nurses caring for patients with dementia exhibiting problematic behaviors was found to be higher when the work environment, including staff allocation, was inadequate [31]. Additionally, sleep disturbances and nocturnal behaviors in patients significantly increased nurses’ stress levels [26,32,34]. The age and work experience of nurses also had an impact on their stress levels [26,35]. Aggressive behaviors in patients affected both subjective and objective burdens on nurses, while ADL (Activities of Daily Living) impairment influenced only the objective burden and did not affect the subjective burden [33]. Providing education for patients and nurses, along with adequate financial and staffing support, improved job satisfaction. Additionally, professional development and workplace atmosphere enhancement contributed to reducing workload-related stress among nurses [29,36].

## 4. Discussion 

This study analyzed domestic and international research on workload-related stress in nurses caring for dementia patients with problematic behaviors by using a literature review approach. The aims of this study were to explore future research directions, propose strategies to reduce nurses’ turnover rates, and provide effective nursing interventions. We analyzed the quantitative results of selected studies, the tools used to measure nurses’ workload-related stress, and the outcomes.

Our findings confirm the steady stream of research conducted annually since 2013. While qualitative studies have been conducted on the burden and stress experienced by nurses caring for patients with dementia, quantitative studies that validate these findings are relatively scarce. The increasing aging population and the prevalence of dementia since 2018 have heightened the global interest in this field [2]. Studies have been conducted primarily in South Korea [25,30,37] and Japan [27,34], with additional research in Jordan, Germany, the Netherlands, and Switzerland, underscoring the worldwide significance of this issue. The findings of this study indicate that dementia-related research is actively being conducted in Asian countries such as South Korea and Japan. This trend appears to be influenced by a combination of social and cultural factors. South Korea and Japan are among the countries with the fastest-aging populations globally, which leads to an increase in the prevalence of dementia and associated challenges [1,2]. In response to this demographic shift, specific welfare policies, such as South Korea’s “National Responsibility for Dementia” initiative, underscore the significance of dementia research and provide essential funding and support for these endeavors [2].

Moreover, the family-centered cultural context prevalent in Asia further accentuates the necessity for research focused on the stress associated with dementia care and related management strategies [40,41]. These cultural characteristics contribute to the heightened demand for studies that address the unique challenges faced by caregivers within these societies.

Therefore, it is imperative to develop culturally tailored intervention strategies and enhance policies and systems that consider the distinctive social and cultural backgrounds of Asian countries. Such efforts are crucial for providing practical support for dementia management and alleviating the work-related stress experienced by nurses. Social support can minimize nurses’ work-related stress and burnout [41]. By addressing these specific needs, the healthcare system can better support both dementia patients and their caregivers, ultimately leading to improved care outcomes.

In this study, workload-related stress in nurses caring for problematic behaviors in dementia patients was measured using terms such as ‘stress’, ‘burden’, and ‘quality of life’ (QoL). The Neuropsychiatric Inventory (NPI) was the most commonly used tool in three studies to assess stress in nurses and caregivers [39]. Five studies used other work-related stress measurement tools. However, diverse stress measurement tools may not accurately confirm the specific stressors faced by nurses caring for patients with problematic dementia [42]. ‘Stress’ and ‘burden’ are terms often used interchangeably to describe psychological states [39], but they may fail to capture the full range of difficulties and psychological challenges experienced by nurses. Therefore, there is a need to develop tools that can more precisely measure these stressors.

Our study confirmed that nurses caring for patients with dementia and problematic behaviors experience significant workload-related stress. This stress arises from the uncertainty and burden caused by patient behavior [12]. Moreover, clinical settings requiring hospitalization for dementia treatment may intensify nurses’ stress because of the demand for advanced medical services and increased responsibilities [43]. All problematic behaviors of patients with dementia have been identified as major stressors for nurses, with aggression being the most prominent stress factor [44]. 

These behaviors increase nurses’ workload-related stress, regardless of the type or cause of dementia [12,45].

Furthermore, research indicates that nurses’ workload-related stress due to the problematic behaviors of patients with dementia negatively impacts nursing performance [45]. Lower nursing performance worsens the quality of care for dementia patients [46], potentially leading to deteriorating patient conditions. Increased workload and job-related stress contribute to high turnover rates among nurses [10], thereby creating a vicious cycle exacerbated by inadequate nursing staff. 

Therefore, effective management and intervention strategies are essential to alleviate nurses’ stress caused by the problematic behaviors of patients with dementia. Understanding and knowledge of these behaviors are crucial because they vary, depending on dementia education and experience [46]. Developing various programs and strategies to reduce nurses’ burdens, decrease job-related stress, improve dementia nursing quality, and prevent nurse turnover is necessary. Furthermore, additional research is needed to objectively evaluate nurses’ workload-related stress caused by problematic behaviors and to develop measures for appropriate staffing levels.

Problematic behaviors in patients with dementia are a primary predictor of nursing workload, regardless of the type of dementia [12,45]. Owing to the unpredictability of dementia, increased workload and feelings of frustration are inevitable, leading nurses to experience overwhelming responsibilities and time constraints [4]. Therefore, computational methods to calculate changes in nursing workload, based on the severity of dementia symptoms, are needed to ensure that nurses can provide adequate care without feeling excessively burdened. 

Also, based on these findings, sufficient nursing staff support is necessary. Dementia’s unpredictable nature and ambiguous communication patterns increase nurses’ workload [47]. Inadequate communication between nurses and patients reduces nurses’ ability to understand the patients’ emotions and needs, thereby increasing the burden of dementia care [48]. Nurses often feel pressured by high workloads and time constraints, frequently experiencing a lack of sufficient time to complete their duties. Problematic behaviors exacerbate this burden, and especially, vague communication further complicates their tasks [49,50,51]. Emotional exhaustion from dealing with problematic behaviors can threaten nurses’ professional identity, leading to significant stress [51,52]. Declined communication with patients can worsen the behavioral and psychological symptoms of dementia (BPSD), further increasing nurses’ workload [53]. Therefore, communication training can be beneficial. Counseling and mindfulness programs are necessary to reduce stress. Adequate material and human resources can alleviate job-related burdens [47], highlighting the need for sufficient nursing staff support.

There is a significant demand for education on managing problematic behaviors in dementia patients [44]. Adequate training enhances nurses’ ability to assess patients’ needs and respond appropriately, reducing the challenges they face in their work [54]. This highlights the importance of developing educational programs focused on understanding and addressing dementia-related behaviors, which can alleviate the burden on nurses and improve patient care.

## 5. Limitations

This study primarily utilized quantitative methods to understand the current research trends, with a focus on workload measurement tools. One limitation of this approach is that it may restrict the generalizability of the findings due to the specific period during which the data were collected. The exclusive use of quantitative data may not fully capture the complexities and variations in the workload experienced by nurses caring for dementia patients. To address these limitations, future research should incorporate qualitative methods. Qualitative research can provide a more comprehensive and nuanced understanding of the topic, offering insights into the subjective experiences and challenges faced by nurses. Additionally, future studies should aim to examine the workload levels according to different patient characteristics. This approach will help in developing tailored interventions and policies that better support the diverse needs of dementia patients and their caregivers, ultimately leading to improved care outcomes and reduced workload-related stress for nurses.

## 6. Conclusions

This study systematically reviewed domestic and international research on workload-related stress among nurses caring for dementia patients with problematic behaviors. The aim of this study was to identify research trends, propose future research directions, and suggest strategies to reduce workload-related stress and turnover rates, while enhancing effective nursing interventions. The findings confirm that nurses caring for patients with dementia and problematic behaviors experience high levels of workload-related stress. These problematic behaviors significantly contribute to stress among nurses, with aggression identified as a major stress factor. Workload-related stress can affect nurses’ personal health and patient safety. Therefore, nursing interventions to manage problematic behaviors in patients with dementia are crucial.

To address these challenges, it is essential to develop policies and provide support systems that prioritize the well-being of nurses. Educational programs targeting nurses to effectively manage these behaviors and reduce stress are essential. Additionally, there is a need for policy development that includes adequate staffing, mental health support, and resources for continuous professional development. By implementing these strategies, healthcare systems can enhance the quality of care for dementia patients while reducing stress and turnover rates among nurses.

## Figures and Tables

**Figure 1 healthcare-12-01893-f001:**
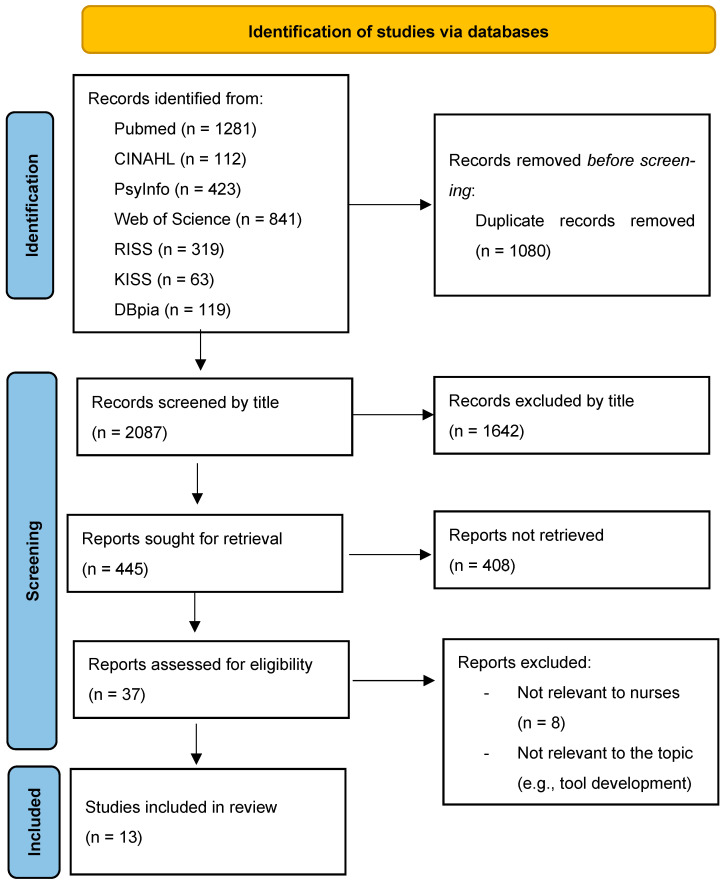
Flow chart of the literature search process. Domestic research databases. RISS (Research Information Sharing Service); KISS (Korean Studies Information Service System); DBpia (Digital Bibliographic Information Access).

**Table 1 healthcare-12-01893-t001:** Summary of the included studies.

Reference No	Aims	Population/Region/Year	Outcome Measurements	Outcomes
[26]	To understand the impact of behavioral and psychological symptoms of dementia (BPSD) in nursing homes on the burden experienced by caregivers	145 dementia patients and 145 caregivers in nursing facilities/South Korea/2013	Mini-Mental State Exam (K-MMSE), Physical function measurement tool (PADL/IADL), Cohen-Mansfield Agitation Inventory (CMAI)	High burden in caregivers linked to specific dementia symptoms, younger caregiver age, shorter patient stay, and higher BPSD scores
[27]	To investigate job-related stress characteristics and structure among dementia nurses (PDNs) to provide mental health management for nurses	244 nurses (63 PDNs, 181 other psychiatric nurses)/Japan/2014	Comparison of stress levels and factors	Higher stress levels in dementia nurses
[28]	To investigate the relationship between individual neuropsychiatric symptoms, frequency, and severity and care staff distress	432 dementia patients in 17 nursing homes/The Netherland/2014	Care staff distress measured by NPI	Distress levels correlated with specific neuropsychiatric symptoms
[29]	To evaluate the effectiveness of an educational program for managing BPSD	204 staff and 187 dementia residents in 16 aged care facilities/New Zealand/2015	Effectiveness of the educational program	Improved staff knowledge and reduced stress
[30]	To compare BPSD characteristics and caregiver burden based on dementia type	214 residents (131 Alzheimer’s type, 83 vascular dementia)/South Korea/2017	Comparison of BPSD characteristics and burden levels	Higher incidence of apathy/indifference, significant differences in anxiety, elation/euphoria, and irritability/instability between Alzheimer’s and vascular dementia groups
[31]	To explore the association between unit type and care worker stress	3922 care workers in 156 Swiss nursing homes/Swiss/2017	Analysis of stress factors	Stress levels associated with unit type and work environment characteristics
[32]	To compare distress levels in caregivers of young-onset dementia (YOD) and late-onset dementia (LOD) patients	382 caregivers of YOD and 261 caregivers of LOD/Dutch/2018	Comparison of distress levels	Higher distress levels in YOD caregivers
[33]	To explore patient factors related to nursing care burden for dementia patients	55 institutionalized dementia patients/Canada/2018	Exploration of care burden factors	Care burden correlated with severity of cognitive impairment and behavioral symptoms
[34]	To identify predictive factors associated with psychological distress in caregivers of dementia patients	1437 caregivers of dementia patients/Japan/2018	Measurement of caregiver distress based on BPSD subcategories	High psychological distress linked to severe BPSD and caregiver burden
[35]	To predict burden areas for caregivers of frontotemporal degeneration (FTD) dementia patients	674 FTD caregivers/USA/2019	Assessment of caregiver burden	Increased neuropsychiatric symptoms linked to higher caregiver burden
[36]	To identify predictors of work stress among nurses caring for elderly patients in acute care settings in Amman, Jordan	485 nurses (public hospitals, private hospitals, healthcare centers)/Jordan/2019	Identification of stress predictors	High workload, low support, and emotional demands as key stress predictors
[37]	To investigate the relationship between BPSD nursing burden and dementia care performance to reduce burden and improve care quality	248 nurses in 27 nursing hospitals/South Korea/2020	Assessment of nursing burden and care performance	BPSD nursing burden significantly negatively correlated with dementia care performance
[38]	To evaluate the effectiveness of mindfulness-based interventions in reducing compassion fatigue and burnout among nurse caregivers	Nurse caregivers of institutionalized elderly persons with dementia/Spain/2022	Evaluation of intervention effectiveness	Reduced compassion fatigue and burnout, improved caregiver well-being

**Table 2 healthcare-12-01893-t002:** Study design of the included studies (N = 13).

Study Design	Cross-Sectional Study	Quasi-Experimental
n (%)	11 (84.6)	2 (15.4)

**Table 3 healthcare-12-01893-t003:** The measurement tools for work-related stress in nurses.

Variable	No	Stress Measure
Stress	28, 31, 32	Neuropsychiatric Inventory Questionnaire (NPI)
	30	Strains in Dementia Care Scale (SDCS)
	27	Psychiatric Nurse Job Stress Scale (PNJSS)
	27	Brief Job Stress Questionnaire (BJSQ)
	32	Health Profession Stress Inventory (HPSI)
	35	Occupational Psychological Distress (K6)
Burden	26	Professional Caregiver Burden Index (PCBI)
	38	Care Burden of the Dementia Disturbing Behaviors Scale
	34	Modified Nursing Care Assessment Scale (M-NCAS)
	37	Quality Work Competence (QWC)
	36	Zarit Burden Inventory (ZBI)
QoL	29	Professional Quality of Life Scale (ProQoL)
	37	Nurse’s Working Life Questionnaire

## Data Availability

Data sharing is not applicable. No new data were created or analyzed in this study.

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
