# Peer review of "Workload-Related Issues among Nurses Caring for Patients with Behavioral and Psychological Symptoms of Dementia: A Scoping Review"

_healthcare, 2024, doi:10.3390/healthcare12181893_

Round 1

Reviewer 1 Report

Comments and Suggestions for Authors

I recommend adding to the affiliation of the authors the city where the university is located + the state/country.  

Manuscript is quite clear, relevant to the field a presented in a quite well-structured manner. It has all part for review article according the Guide for Authors. A minor correction of the aim of study and Purposes of study is needed and to add information in the part material and Methods.   

The all manuscript is written in an appropriate way, quality is a quite high. 

The abstract is too short, it is appropriate to add the methdology (to add search strategy – key words, selected databases and analysis of papers/studies) and results. In the results to add information about number of papers were included in the scoping review. I recommend reevaluating the keywords in the context of the title of paper. The keywords nurse stress, work-related stress remove and replase them withother words in the context of title of the manuscript.        

The part Introduction is original and relevant according the content in the context of the title of this manuscript. The aim (in Introduction) and in the and Purposes of the study are similar due to the content of the text, it is not needed to repeated.  

The part Material and Methods is original and relevant. The methodology is quite clearly and comprehensibly described. I suggest to add the information about paritcipats – in the study were included hospitalized patients?, living in personal residential house?, in home with the family? Or?  It was dependent at age of participatnt or stage of dementia disease?

The Results are interpreted in well quality. It is quite clear part of manuscript. I suggest little formal correction of title in Tables nomber of 2, 3 and 4 – see text below.      

Limitation of the present study are relevant.

Discussion is written in appropriate way.    

Conclusions are written clearly and comprehensibly. The content of this part of manuscript is a summary of the study results and it consistent. The conclusions are interesting for the readership of this journal (authors wrote about domestic and international research).    

All references (in the number of 45) are appropriate in the manuscript. It is needed to check formal notation of the all references in the paper manuscripts according the Guide for Authors.   

In manuscript are included one figure and four tables. I propose a minor modification of the designation of Figure 1. : PRISMA flow diagram of ... In Figure 1 were used abbreviatons as RISS, KISS, SBpia without explanation, presase to add information about this in the paper manuscript. I suggest a small formal modification of the numerical notation of the year of publication of the study in Table 2- currently it is in two lines below one another, which worsens understanding. In the Table 1, the name of the table label is in bold (it is not in two others). I suggest a small content correction of title Table 3 – not Study design but Study design of included studies (N = 13). I suggest tu correct the title of Table 4 – the content is about the measurement tools of work-related stress in nurses?, please to modified the title according the content of this table.          

All (figure and tables) are appropriate according the content and they are properly show the data. 

The ethics statements is not applicable, it is in part of Istituinal Review Board Statement.

Authors declared no external funding and no conflict of interest, too.

According me, the English language is appropriate and undestandable, though I´m not a native speaker in English. 

I think, that this manuscript/paper will be attract a wide readership for more people/readers.    

Overall recommendation for next processing stage of the manuscript: accept after minor revisions. 

Author Response

Dear Reviewer,

Thank you for considering our article for publication in Healthcare. We sincerely appreciate the reviewer’s valuable feedback on our manuscript and are happy to make the necessary changes. Below are our responses to each of the comments provided:

Comment 1: I recommend adding the city where the university is located and the state/country to the affiliation of the authors.

  • Response: In response to your suggestion, we have added the city and country to the authors’ affiliations.

Comment 2: The manuscript is quite clear, relevant to the field, and well-structured. It contains all parts required for a review article according to the Guide for Authors. A minor correction of the study aim and purpose is needed, along with additional information in the Materials and Methods section. The abstract is too short; it is appropriate to add methodology (search strategy, keywords, selected databases, analysis) and results, including the number of papers included in the scoping review. I also recommend re-evaluating the keywords in relation to the title, replacing terms like "nurse stress" and "work-related stress" with others that better reflect the manuscript’s title.

  • Response: We have revised the abstract to include methodology and reworked the keywords as suggested. In addition, we removed the repetitive parts in the introduction that duplicated the study aim and purpose, providing clearer distinctions.

Comment 3: The Materials and Methods section is original and relevant, but more information is needed about the participants: Were the patients hospitalized? Living in a personal residence or with family? Was participation dependent on age or dementia stage?

  • Response: The participants included in the analyzed studies were all Registered Nurses (RNs). Studies that did not focus on RNs were excluded. As the patients in these studies were being cared for in hospitals or centers by nurses, differences in the patients' living environments were not considered. Detailed information has been provided in Table 1 under the “Population” section.

Comment 4: All references (45 in total) are appropriate. Please check the formal notation of references according to the Guide for Authors. A minor adjustment is needed for Figure 1, where abbreviations (RISS, KISS, DBpia) are used without explanation. Also, correct the numerical formatting in Table 2 and make slight changes to the titles of Tables 3 and 4.

  • Response: We have corrected the reference formatting according to the Guide for Authors. Additionally, we added explanations for RISS, KISS, and DBpia in the figure, and modified the formatting and titles of the tables as requested.

Reviewer 2 Report

Comments and Suggestions for Authors

The work situation for caring staff dealing with demented patients who have difficult behaviors is an important subject for researchers. I must say, however, that the way in which the authors decided to make their literature search has limited the likelihood of finding interesting avenues for dealing with the problems. In addition, the relatively short search window, 10 years, excludes some groundbreaking studies published before that period, which is a point that the authors are making themselves.

I can understand that the authors are focusing on institutional staff rather than relatives taking care of this groups of patients at home, but this means that an important part of the total spectrum has been lost.

The limitation of study selection also excludes the literature on physiological outcomes, such as immunological parameters.

Another avenue that the authors have lost is the one dealing with alternative ways of creating improved social interaction between staff and this groups of patients. I think of singing and other kinds of musical activities, dancing and visual arts for creating dialogue etc. There are quantitative studies which show that such activities can indeed be effective.

This study has been technically well conducted and the formal aspects are good. However, I do not feel that the study contributes significantly to the literature

Author Response

Dear Reviewer,

Thank you for considering our article for publication in Healthcare. We sincerely appreciate the reviewer’s valuable feedback on our manuscript and are happy to make the necessary changes. Below are our responses to each of the comments provided:

Comment 1: The work situation for caring staff dealing with dementia patients who have difficult behaviors is an important subject for researchers. However, I must say that the way the authors decided to make their literature search has limited the likelihood of finding interesting avenues for dealing with these problems. In addition, the relatively short search window, 10 years, excludes some groundbreaking studies published before that period, which is a point that the authors themselves acknowledge.
I can understand that the authors are focusing on institutional staff rather than relatives caring for these groups of patients at home, but this means an important part of the total spectrum has been lost.
The limitation of the study selection also excludes the literature on physiological outcomes, such as immunological parameters.
Another avenue that the authors have lost is the one dealing with alternative ways of creating improved social interaction between staff and these groups of patients. I think of singing and other kinds of musical activities, dancing, and visual arts for creating dialogue, etc. There are quantitative studies that show such activities can indeed be effective.
This study has been technically well conducted, and the formal aspects are good. However, I do not feel that the study contributes significantly to the literature.

  • Response: Thank you very much for your valuable and insightful feedback. We appreciate your thorough and critical comments. We agree with your point about the relatively short search window of 10 years. Our focus on studies from the last 10 years was intended to capture the most recent trends and interventions faced by nurses caring for dementia patients. However, as you pointed out, we recognize that important studies published before this period may have been excluded.
    Our decision to focus on institutional nurses rather than family caregivers was based on the fact that the challenges faced by these groups may differ. However, we acknowledge that this focus could have excluded significant findings related to home caregivers, as you mentioned. We will consider including both institutional and home care settings in future research.
    Additionally, we appreciate your suggestion to consider physiological outcomes, such as immunological parameters, and alternative methods of improving social interaction between staff and patients, including music, art, and dance. We recognize the importance of these factors and will take them into account in future studies. While we believe this review provides important insights into the issues faced by institutional nurses, we will strive to improve the scope and quality of our research based on your valuable suggestions.
    Once again, we thank you for your thoughtful review and constructive feedback. We will incorporate your recommendations to further enhance the quality and scope of our manuscript.

Reviewer 3 Report

Comments and Suggestions for Authors

A scoping review of seven databases over January 1, 2013, and December 31, 2023, of 3,167 articles to determine workload-related issues among nurses caring for dementia patients exhibiting problematic behaviors to find the results and make suggestions, particularly concerning Korean nurses.

The review is well-conceived and executed following the required PRISMA procedures. The results are well-documented, with a clear and direct writing style and English usage. 

There are 45 references; only nine are to research published since 2020. The gold standard for scientific research is that references are to research from the last five years. Other than those articles as part of the scoping review, either substitute or support outdated citations with current research published since 2020.

The authors have used several referencing styles. Please switch all the references to the MDPI style required by Healthcare.

Line by line suggested edits

10-16 As a scoping review, please adjust the Abstract to include background, objectives, eligibility criteria, sources of evidence, charting methods, results, and conclusions relating to the review questions and objectives. Once completing the Abstract according to the requirement of a scoping review, please adjust item 2 on the Preferred Reporting Items for Systematic Reviews and Meta-Analyses extension for Scoping Reviews (PRISMA-ScR) Checklist to provide the page number.

31-33 Please cite a current reference to support this claim.

47 Please define burnout and cite the seminal work in this area, plus current research regarding nurses and burnout—especially concerning burnout resulting from care for dementia patients.

55 Please provide examples of the challenges and burdens experienced by nurses when caring for dementia patients supported by citations to current references.

66 Change to “study is to”.

69 Change to “study is to”.

74 Change to “search is to”.

76 Change to “study follows”.

76-80 Please explain the choice of this framework and provide citations to research demonstrating its current use by other researchers.

88 Change to “follows”.

90 Change to “targets”.

90 Please explain why the authors conducted the scoping review over the last 10 years rather than the five years and cite other scoping reviews of the topic that have also done so as support for this decision. 

90-91 Change to “published from January 1, 2013 to December 31, 2023 that”.

112-113 Please provide additional detail in the final right-hand box of the figure regarding why the reports were excluded related to the parameters searched.

116 Change to “follows the scoping ”. Please explain why the authors follow the recommendations of Armstrong from 2011 when the requirements for scoping reviews were reconsidered and republished in 2020.

132-133 Please include a column indicating the year of publication. Once completed, Table 2 is unnecessary. To follow the requirements for tables of the journal, please use the center justification of the entries in the columns rather than the left justification.

138-147 Please cite studies using square brackets, not round ones.

141-142 Please change the font to correspond with the rest of the manuscript.

147-152 Please provide a seminal citation for the NPI.

160-161 Please create horizontal lines dividing Stress, Burden, and QoL in Table 4 to differentiate the three variables visually.

162 Change to “reveals.

162-148 Please cite studies using square brackets, not round ones.

189 Change to “analyzes”.

191 Change “were” to “are”.

193 Change to “analyze”.

200 Please provide the citation numbers rather than the number of studies.

226 Change to “confirms”. Change “who problematic” to “with problematic”.

251-254 Please cite current research to support this claim.

259-260 Please cite current research to support this claim.

275 Change to “utilizes”.

289 Change to “This study represents a scoping review of domestic”.

Comments on the Quality of English Language

Suggested changes to the quality of English usage are in the Comments and Suggestions for Authors. 

Author Response

Dear Reviewer,

Thank you for considering our article for publication in Healthcare. We sincerely appreciate the reviewer’s valuable feedback on our manuscript and are happy to make the necessary changes. Below are our responses to each of the comments provided:

Comment 1: Only nine out of 45 references are from research published since 2020. Please replace or support outdated references with more recent studies. Additionally, the references should be formatted according to the MDPI style required by Healthcare.

  • Response: We have replaced and supplemented outdated references with more recent studies and reformatted the references according to the MDPI style.

Comment 2: Please make changes to the following line numbers (66, 69, 74, 76, 88, 90, 116, 162, 189, 191, 193, 226, 275).

  • Response: We have made all the suggested changes to these lines.

Comment 3: The abstract needs to follow the scoping review structure, including background, objectives, eligibility criteria, sources of evidence, charting methods, results, and conclusions. Adjust item 2 of the PRISMA-ScR checklist accordingly.

  • Response: We have revised the abstract and adjusted the PRISMA-ScR checklist as requested. Changes are marked in red on pages 1-2.

Comment 4: Please explain the choice of the framework and provide citations to research demonstrating its current use by other researchers.

  • Response: We have revised the manuscript to explain the framework and included additional citations. These changes are marked in red on pages 2-4.

Comment 5: Please explain why the authors conducted the scoping review over the last ten years instead of five.

  • Response: We chose to review studies from the past ten years to reflect the most recent trends and interventions for nurses caring for dementia patients. Additionally, we aimed to address a gap in research related to work-related issues for institutional nurses, where fewer studies have been conducted. Similar examples of extended search windows exist, and we plan to refine the scope in future studies to focus on specific timeframes.

Comment 6: Provide additional details in the final right-hand box of the flowchart regarding why reports were excluded.

  • Response: We have updated the flowchart with the requested details.

Comment 7: Use square brackets instead of round ones for citations, adjust the font in lines 141-142, and provide a seminal citation for the NPI.

  • Response: We have made the necessary adjustments and marked them in red.

Round 2

Reviewer 2 Report

Comments and Suggestions for Authors

The authors provide adequate explanations 

Comments on the Quality of English Language

I discovered minor errors, for instance one on line 238.

But these are small problems

Author Response

Dear Reviewer 2,

Comment: I discovered minor errors, for instance one on line 238.

Response:  Thank you for your feedback.

I have reviewed line 238 and identified the error. The typo has now been corrected.

I appreciate your attention to detail.

Reviewer 3 Report

Comments and Suggestions for Authors

Thank you to the authors for their changes to the manuscript. All have improved it. There are more that need attention.

Although the authors have improved the referencing style, they have used more than one style. Furthermore, they have not followed the instructions for referencing accurately. Please modify all the references to the MDPI style required by Healthcare. Please see https://www.mdpi.com/journal/healthcare/instructions.

Line by line suggested edits

42 Please find a supporting citation for 3 published since 2020.

44 Please find a supporting citation for 4 published since 2020.

51 Please find a supporting citation for 7 published since 2020.

55 Please find a supporting citation for 9 published since 2020.

57 Please find a supporting citation for 10 published since 2020.

60 Please find a supporting citation for 12 and 13 published since 2020.

61 Please define burnout in the text and cite the seminal work.

65 Please find a supporting citation for 19 published since 2020.

67 Please find a supporting citation for 17 published since 2020.

92 “covers Stage”—covers Stage what?

93 Indicate whether a review protocol exists

94-99 The research question belongs in the Introduction, not the Materials and Methods. Please move this section to the end of the Introduction.

103-104 Please state, in the text, why the search is over ten years rather than 5.

125-126 Please expand the size of Figure 1 to the entire page.

128 Please change the font to correspond to the rest of the manuscript.

129-134 In the previous review, the authors were asked to explain why they followed the recommendations of Armstrong from 2011 when the requirements for scoping reviews were reconsidered and republished in 2020. Rather than explain, they have followed the recommendations of Arksey and O’Malley from 2005. The authors need to state in the text why they have followed their particular recommendations for scoping reviews.

147-148 Please change “Population/Region/Years” to “Population/Region/Year”.

167 Please find a supporting citation for 37 published since 2020 and change the font to match the rest of the manuscript.

174 Please change the font to match the rest of the manuscript.

175 Please change the font to match the rest of the manuscript.

176 Please change “reveals” to the font that matches the rest of the manuscript.

214 Please find a supporting citation of research published since 2020 to support citations 25 and 32.

232  Please find a supporting citation of research published since 2020 to support citation 39. Please change the font of 39 to match the rest of the manuscript.

238, 242  Please find a supporting citation of research published since 2020 to support citation 37. Please change the font of the second instance of 37 to match the rest of the manuscript.

241 Please change the font of 40 to match the rest of the manuscript.

247 Please find a supporting citation of research published since 2020 to support citation 12.

252 Please find a supporting citation of research published since 2020 to support citations 12 and 43.

275 Please find a supporting citation of research published since 2020 to support citation 10.

268 Please find a supporting citation of research published since 2020 to support citations 12 and 43. Please change the font of both citations to match the rest of the manuscript.

270 Please find a supporting citation of research published since 2020 to support citation 4.

276 Please find a supporting citation of research published since 2020 to support citation 45.

282 Please find a supporting citation of research published since 2020 to support citations 49 and 50. Please also put a space before the first bracket.

284 Please find a supporting citation of research published since 2020 to support citation 51. Please also put a space before the first bracket.

289 284 Please find a supporting citation of research published since 2020 to support citation 51. Please also put a space before the first bracket.

289 Please find a supporting citation of research published since 2020 to support citation 42. Please also put a space before the first bracket.

290 Please put a space before the first bracket.

Preferred Reporting Items for Systematic reviews and Meta-Analyses extension for

Scoping Reviews (PRISMA-ScR) Checklist

Please enter the page number of the Abstract.

Please state where the authors have indicated whether a review protocol exists.

All of the page numbers listed are incorrect. Please revise this document.

Comments on the Quality of English Language

Suggested changes to the quality of English usage are in the Comments and Suggestions for Authors. 

Author Response

Response to Reviewer Comments

Dear Reviewer 3,

Thank you for considering our article for publication in Healthcare. We sincerely appreciate the reviewer’s valuable feedback on our manuscript and are happy to make the necessary changes. We have highlighted the revised sections in red and blue. We have replaced all references with the most recent literature published after 2020, as much as possible. Due to the addition of new references, the numbering has changed (e.g., references 10 and 11 were added, so the previous reference 10 is now 12).
The response letter has been written line by line, and the revised sections have been highlighted in red and blue for emphasis.

Below are our responses to each of the comments provided

Reviewer 3

  1. Although the authors have improved the referencing style, they have used more than one style. Furthermore, they have not followed the instructions for referencing accurately. Please modify all the references to the MDPI style required by Please see https://www.mdpi.com/journal/healthcare/instructions.

Reply: Reformatted the references according to the MDPI style.

Line by line suggested edits

  1. 42 Please find a supporting citation for 3 published since 2020.

Reply: I have incorporated references to the most recent research published since 2020.

  1. 44 Please find a supporting citation for 4 published since 2020.

Reply: I have incorporated references to the most recent research published since 2020.

  1. 51 Please find a supporting citation for 7 published since 2020.

Reply: Thank you for your thoughtful suggestion. The cited literature plays a crucial role in contextualizing the research and emphasizing its necessity. Unfortunately, I could not identify a more recent source published after 2020 that serves the same purpose.

  1. 55 Please find a supporting citation for 9 published since 2020.

Reply: I have incorporated references to the most recent research published since 2020.

  1. 57 Please find a supporting citation for 10 published since 2020.

Reply: The cited literature plays a crucial role in contextualizing the research and emphasizing its necessity. Unfortunately, I could not identify a more recent source published after 2020 that serves the same purpose.

  1. 60 Please find a supporting citation for 12 and 13 published since 2020.

Reply: I have incorporated references to the most recent research published since 2020.

  1. 61 Please define burnout in the text and cite the seminal work.

Reply: I have incorporated a definition of burnout into the text, drawing upon the seminal work cited in reference 18.

  1. 65 Please find a supporting citation for 19 published since 2020.

Reply: . The citation for reference 19 has not been replaced with a study published after 2020, as the original reference is critical for supporting the specific context of the discussion

  1. 67 Please find a supporting citation for 17 published since 2020.

Reply: The cited literature plays a crucial role in contextualizing the research and emphasizing its necessity. Unfortunately, I could not identify a more recent source published after 2020 that serves the same purpose.

  1. 92 “covers Stage”—covers Stage what?

Reply: The phrase was intended to refer to "Stage 5." I have corrected the text to include the missing number.

  1. 93 Indicate whether a review protocol exists

Reply: I have provided a detailed description of the review protocol in the Methods section on pages 2, 3.

  1. 94-99 The research question belongs in the Introduction, not the Materials and Methods. Please move this section to the end of the Introduction.

Reply: The necessary revisions have been made accordingly.

  1. 103-104 Please state, in the text, why the search is over ten years rather than 5.

Reply: We selected a 10-year period because it is often recommended as an appropriate time frame that reflects recent research trends while ensuring sufficient research data on the topic. We have provided a detailed explanation of this rationale in Section 2.2 on page 3.

  1. 125-126 Please expand the size of Figure 1 to the entire page.

Reply: The necessary revisions have been made accordingly.

  1. 128 Please change the font to correspond to the rest of the manuscript.

Reply: The necessary revisions have been made accordingly.

  1. 129-134 In the previous review, the authors were asked to explain why they followed the recommendations of Armstrong from 2011 when the requirements for scoping reviews were reconsidered and republished in 2020. Rather than explain, they have followed the recommendations of Arksey and O’Malley from 2005. The authors need to state in the text why they have followed their particular recommendations for scoping reviews.

Reply: Thank you for your careful review. We followed the Arksey and O’Malley (2005) framework because it is particularly well-suited for scoping reviews aimed at mapping available evidence, identifying knowledge gaps, clarifying key concepts, and examining how previous studies on related topics were conducted. While we acknowledge the updated guidelines from 2020, the Arksey and O'Malley framework remains relevant and widely used for the objectives of this review. We have now included an explanation in the text to clarify our choice.

  1. 147-148 Please change “Population/Region/Years” to “Population/Region/Year”.

Reply: The necessary revisions have been made accordingly.

  1. 167 Please find a supporting citation for 37 published since 2020 and change the font to match the rest of the manuscript.

Reply: Thank you for your suggestion. The original journal was referenced to explain the concept of NPI, and therefore, we have not replaced it with a more recent source. Additionally, the reference previously listed as 37 has been updated to 39 with an additional citation.

  1. 174 Please change the font to match the rest of the manuscript.

Reply: The necessary revisions have been made accordingly.

  1. 175 Please change the font to match the rest of the manuscript.

Reply: The necessary revisions have been made accordingly.

  1. 176 Please change “reveals” to the font that matches the rest of the manuscript.

Reply: The necessary revisions have been made accordingly.

  1. 214 Please find a supporting citation of research published since 2020 to support citations 25 and 32.

Reply: References 25, 32 were key papers used in the literature analysis, have not been replaced. The addition of new citations, these references No 27, 34 respectively.

  1. 232  Please find a supporting citation of research published since 2020 to support citation 39. Please change the font of 39 to match the rest of the manuscript.

Reply: The cited literature plays a crucial role in contextualizing the research and emphasizing its necessity. Unfortunately, I could not identify a more recent source published after 2020 that serves the same purpose.

  1. 238, 242  Please find a supporting citation of research published since 2020 to support citation 37. Please change the font of the second instance of 37 to match the rest of the manuscript.

Reply: The original journal was referenced to explain the concept of NPI, and therefore, we have not replaced it with a more recent source. Additionally, the reference previously listed as 37 has been updated to 39 with an additional citation.

  1. 241 Please change the font of 40 to match the rest of the manuscript.

Reply: The necessary revisions have been made accordingly.

  1. 247 Please find a supporting citation of research published since 2020 to support citation 12.

Reply: I have incorporated references to the most recent research published since 2020.

  1. 252 Please find a supporting citation of research published since 2020 to support citations 12 and 43.

Reply: I have incorporated references to the most recent research published since 2020.

Please note that reference 43 has now been updated to reference 45 following the addition of new citations.

  1. 275 Please find a supporting citation of research published since 2020 to support citation 10.

Reply: The cited literature plays a crucial role in contextualizing the research and emphasizing its necessity. Unfortunately, I could not identify a more recent source published after 2020 that serves the same purpose.

  1. 268 Please find a supporting citation of research published since 2020 to support citations 12 and 43. Please change the font of both citations to match the rest of the manuscript.

Reply: I have incorporated references to the most recent research published since 2020.

 Please note that reference 43 has now been updated to reference 45 following the addition of new citations.

  1. 270 Please find a supporting citation of research published since 2020 to support citation 4.

Reply: I have incorporated references to the most recent research published since 2020.

  1. 276 Please find a supporting citation of research published since 2020 to support citation 45.

 Reply: The cited literature plays a crucial role in contextualizing the research and emphasizing its necessity. Unfortunately, I could not identify a more recent source published after 2020 that serves the same purpose.

Please note that reference 45 has now been updated to reference 47 following the addition of new citations

  1. 282 Please find a supporting citation of research published since 2020 to support citations 49 and 50. Please also put a space before the first bracket.

Reply: The cited literature plays a crucial role in contextualizing the research and emphasizing its necessity. Unfortunately, I could not identify a more recent source published after 2020 that serves the same purpose.

Please note that reference 49,50 has now been updated to reference 51,52 following the addition of new citations

  1. 284 Please find a supporting citation of research published since 2020 to support citation 51. Please also put a space before the first bracket.

Reply: I have incorporated references to the most recent research published since 2020.

Please note that reference 51 has now been updated to reference 53 following the addition of new citations

  1. 289 Please find a supporting citation of research published since 2020 to support citation 42. Please also put a space before the first bracket.

Reply: I have incorporated references to the most recent research published since 2020.

  1. 290 Please put a space before the first bracket.

Reply : The necessary revisions have been made accordingly.

Preferred Reporting Items for Systematic reviews and Meta-Analyses extension for Scoping Reviews (PRISMA-ScR) Checklist

  1. Please enter the page number of the Abstract.

Reply: The necessary revisions have been made accordingly. The revised manuscript attached.

  1. Please state where the authors have indicated whether a review protocol exists.

Reply: I have made the necessary additions to the Methods section on page 3.

  1. All of the page numbers listed are incorrect. Please revise this document.

Reply: The necessary revisions have been made accordingly.